# The Anion Channel TMEM16a/Ano1 Modulates CFTR Activity, but Does Not Function as an Apical Anion Channel in Colonic Epithelium from Cystic Fibrosis Patients and Healthy Individuals

**DOI:** 10.3390/ijms241814214

**Published:** 2023-09-18

**Authors:** Azam Salari, Renjie Xiu, Mahdi Amiri, Sophia Theres Pallenberg, Rainer Schreiber, Anna-Maria Dittrich, Burkhard Tümmler, Karl Kunzelmann, Ursula Seidler

**Affiliations:** 1Department of Gastroenterology, Hannover Medical School, 30625 Hannover, Germany; salari.azam@mh-hannover.de (A.S.); xiu.renjie@mh-hannover.de (R.X.); amiri.mahdi@mh-hannover.de (M.A.); 2Department of Pediatric Pneumonology, Allergology and Neonatology, Hannover Medical School, 30625 Hannover, Germanydittrich.anna-maria@mh-hannover.de (A.-M.D.);; 3Institute of Physiology, University of Regensburg, 93040 Regensburg, Germany; rainer.schreiber@vkl.uni-regensburg.de (R.S.); karl.kunzelmann@ur.de (K.K.)

**Keywords:** cystic fibrosis, human colon, enteroids, colonoids, rectal organoids, Ussing chamber, TMEM16a, calcium-activated, purinergic receptors, fluid secretion, anion conductance, chloride

## Abstract

Studies in human colonic cell lines and murine intestine suggest the presence of a Ca^2+^-activated anion channel, presumably TMEM16a. Is there a potential for fluid secretion in patients with severe cystic fibrosis transmembrane conductance regulator (*CFTR*) mutations by activating this alternative pathway? Two-dimensional nondifferentiated colonoid–myofibroblast cocultures resembling transit amplifying/progenitor (TA/PE) cells, as well as differentiated monolayer (DM) cultures resembling near-surface cells, were established from both healthy controls (HLs) and patients with severe functional defects in the *CFTR* gene (PwCF). F508del mutant and CFTR knockout (null) mice ileal and colonic mucosa was also studied. HL TA/PE monolayers displayed a robust short-circuit current response (ΔI_eq_) to UTP (100 µM), forskolin (Fsk, 10 µM) and carbachol (CCH, 100 µM), while ΔI_eq_ was much smaller in differentiated monolayers. The selective TMEM16a inhibitor Ani9 (up to 30 µM) did not alter the response to luminal UTP, significantly decreased Fsk-induced ΔI_eq_, and significantly increased CCH-induced ΔI_eq_ in HL TA/PE colonoid monolayers. The PwCF TA/PE and the PwCF differentiated monolayers displayed negligible agonist-induced ΔI_eq_, without a significant effect of Ani9. When TMEM16a was localized in intracellular structures, a staining in the apical membrane was not detected. TMEM16a is highly expressed in human colonoid monolayers resembling transit amplifying cells of the colonic cryptal neck zone, from both HL and PwCF. While it may play a role in modulating agonist-induced CFTR-mediated anion currents, it is not localized in the apical membrane, and it has no function as an apical anion channel in cystic fibrosis (CF) and healthy human colonic epithelium.

## 1. Introduction

The colonic epithelium secretes anions and fluid to flush the cryptal lumen, thereby removing pathogens, allowing mucus hydration and expansion, bringing antimicrobial peptides to the epithelial surface, and helping to lubricate feces to facilitate evacuation. In the colonic epithelium of patients with severe functional defects in the *CFTR* gene (PwCF) and in *CFTR*-deleted mice, mucus granule expulsion is compromised [1] and expansion, hydration, and removal from the apical membrane is incomplete [2,3], resulting in a sticky mucus gel, clogging of the cryptal lumina, insufficient removal of adherent bacteria, and difficulties with evacuation [4]. Gene expression for non-CFTR anion channels are found in the *CFTR*-deleted murine intestine [5], as well as in biopsies from PwCFs. These channels are being discussed as potential alternative Cl^-^ channels that can, if activated, ameliorate the cystic fibrosis (CF) phenotype [6,7].

Controversy exists regarding the nature of the luminal anion conductance that mediates the Ca^2+^-dependent I_sc_ response in the colon. While several studies of murine colon and some studies of CF patients [8] suggest that the Ca^2+^-dependent stimulation of anion secretion is also mediated via CFTR, although regulated by TMEM16a-depending Ca^2+^ signaling [9], other researchers have provided evidence for a non-CFTR (TMEM16a?)-mediated apical colonic Cl^-^-conductance as a separate entity [10,11,12,13,14].

Our own experiments with CFTR-null mice (with no functional CFTR expressed) and their wild-type (WT) littermates demonstrated a complete absence of change in short-circuit current in response to both Ca^2+^ and cyclic nucleotide-dependent agonists in CFTR-deficient murine colonic epithelium (although an increase in alkaline output due to agonist-mediated NHE3 inhibition was observed). This suggested that the agonists did not stimulate an electrogenic anion channel, which would have resulted in an I_sc_ response [15].

The development of strategies to grow, maintain, and differentiate intestinal epithelium from intestinal biopsies in vitro expand our possibilities to functionally study human native intestinal epithelia from healthy individuals and from patients carrying mutations in the *CFTR* gene [16,17]. Therefore, in order to be able to dissect the electrolyte transport processes in the colonic cryptal base and neck region, we cocultured organoid-derived colonocytes from the transverse colon (on transwell filters) with colonic myofibroblast from the underlying lamina propria (on the bottom of the culture dish) (CM–CE monolayers) [18]. This approach resulted in a rapid increase in transepithelial resistance, but maintained the colonoids in an undifferentiated state, with a gene expression pattern resembling transit-amplifying colonocytes in the lower part of the crypts. These are the enterocytes with particularly high CFTR expression [19]. From the same 3D organoids, differentiated colonocyte monolayers (DMs) were established; they displayed a gene expression pattern resembling cryptal mouth and surface colonocytes. We used the same approach to establish rectal myofibroblast–rectal epithelial cells (RM–RE) and differentiated monolayers from rectal biopsies from patients with cystic fibrosis and from healthy individuals. These organoid-derived monolayers were studied functionally, biochemically, and immunocytochemically for the expression of the Ca^2+^-activated Cl^-^ channel TMEM16a. The ileal and proximal and mid-distal mucosa from mice that either expressed no functional CFTR protein or expressed the F508del mutant were also tested for a response to the same agonists.

## 2. Results

### 2.1. Morphological Features and Expression Profile of Differentiation Markers and Ion Transporters in Nondifferentiated and Differentiated 3D Transverse Colonic and Rectal Organoids from Healthy Volunteers and PwCF

The protocol to generate, expand, and differentiate human colonoids from the transverse colon of healthy volunteers, and the characterization of these organoids, was previously published [18]. Using similar techniques, rectal organoids from PwCF and healthy volunteers, as well as the RM–RE and differentiated monolayers, were established. The images in Figure 1 are from 3D organoid cultures from the transverse colon of a healthy control and from the rectum of a CF patient with a homozygous F508del mutation, in the nondifferentiated and differentiated states (Figure 1A). No lumen is seen in the CF patient’s organoids. Immunocytochemical stainings of 3D organoids from the transverse colon and from the recta of a healthy donor and a homozygous F508del patient are displayed in the nondifferentiated and differentiated states in Figure 1B, with staining for CFTR (green), for the goblet cell marker UEA1 (magenta), for F-actin (white), and for nuclei with DAPI (blue). CFTR was apically expressed in nondifferentiated organoids from the healthy transverse colon and from the rectum (Figure 1B). In contrast, an organellar staining pattern was observed in the rectal organoids of the patient that was homozygous for the F508del CFTR mutant. No CFTR staining was observed in differentiated organoids. While the UEA 1 staining was sparse in the nondifferentiated organoids, the mucus thecae of the goblet cells in the differentiated healthy organoids, which had mostly been expulsed into the lumen, stained strongly. In the differentiated CF organoids, more UEA1 staining was observed, and the mucus thecae were still intracellular. A defect in goblet cell mucus granule expulsion was previously described in small intestinal organoids from CFTR-null mice [1]. The mRNA expression for a panel of genes (see Appendix A) was assessed in the nondifferentiated and differentiated 3D organoids (Figure 1C). The changes in the gene expression pattern between the nondifferentiated and differentiated organoids from the transverse colon and the rectum were similar to those previously described [18]. A stronger decrease in the stem cell and proliferative markers, as well as in the transporter genes involved in anion secretion, and a stronger upregulation in the goblet cell and absorptive enterocyte markers were measured in the differentiated than in the nondifferentiated organoids.

### 2.2. Modeling of Two-Dimensional Transverse Colon Organoid Monolayers with Myofibroblast Coculture

We previously described the establishment of a colonic myofibroblast–colonic epithelial cells (CM–CE) coculture system, which resulted in a rapid increase in the TEER but maintained the colonocytes in a nondifferentiated state [18]. The expression profile of the organoid monolayer was consistent with a cell layer resembling the transit amplifying/progenitor (TA/PE) cells in the lower part of the colonic crypt. In the following paragraphs, to differentiate between cocultures from the transverse colon and from the rectum, we use the terms colonic myofibroblast-colonic epithelial cells (CM–CE) and rectal myofibroblasts-rectal epithelial cells (RM–RE). We assessed the expression of additional genes that are relevant to the present study (Figure 2). On the same day of monolayer culture, the CM–CE and RM–RE cocultures displayed, respectively, the highest mRNA expression of TMEM16a compared to colonoid monolayers grown in expansion medium (EM) (but without a myofibroblast monolayer) or to a differentiated monolayer (DM) (results shown later). These CM–CE monolayers also had robust expression levels for other components of the anion secretory machinery (Figure 2). Therefore, we considered them suitable models for studying the localization and potential functional role of TMEM16a as an apical anion channel.

### 2.3. Electrophysiological Assessment of Anion Channel Activity in CM–-CE Monolayers

As shown above, the CM–CE monolayers express both CFTR and TMEM16a. There is an ongoing controversy about the function and cellular localization of TMEM16a in the colon [9,14,20,21], but few data exist from human native colonic tissue. Therefore, the CM–CE monolayers were studied electrophysiologically in the Easymount voltage clamp system. Since it is known that the dipstick electrodes of the EVOM2 voltmeter for tissue culture (World precision instruments, Sarasota, Fl. USA) overestimate the TEER, it was reassuring to find a robust TEER of approx. 800–1000 Ω.cm^−2^ in the Ussing chamber (Appendix A). The purinergic receptor agonist UTP (100 µM, added luminally) elicited a robust but transient ΔI_eq_, while the cAMP agonist forskolin (10 µM) in combination with the phosphodiesterase inhibitor IBMX (both added basolaterally) elicited a stable I_eq_ increase (Figure 3). Carbachol (CCH, 100 µM) induced a very strong but transient increase in ΔI_eq_. Consistent with the response of a tight epithelium to the opening of apical anion channels, the TEER decreased in parallel with the respective ΔI_eq_ (Appendix A). Preincubation of the CM–CE monolayers with 10 µM or 30 µM Ani9, added either luminally or basolaterally (with no difference), did not alter the response to luminal UTP, significantly reduced the response to Fsk + IBMX, and significantly increased the response to CCH (Figure 3A). Ani9 is suggested to be the most potent and specific TMEM16a inhibitor that is currently available commercially [22], and in these concentrations it is reported to fully inhibit TMEM16a [23]. In the differentiated colonoid monolayer, the response to all agonists was strongly reduced, consistent with the strong downregulation of CFTR mRNA and protein expression (Figure 3B).

To better understand the lack of an inhibitory effect of Ani9 on the UTP-induced ΔI_eq_, immunocytochemical (ICC) staining for TMEM16a was performed, using a variety of previously validated anti-TMEM16a antibodies [9,24,25,26]. The ICC staining revealed an intracellular organellar staining pattern (Figure 4). No colocalization with the apical F-actin stain phalloidin was detected.

### 2.4. Electrophysiological Studies in Intestinal Mucosa from F508del Homozygous and WT Mice

We first repeated experiments similar to those previously published [16], in the CFTR-null (with no CFTR protein at all) small intestine and large intestine. Neither luminal UTP nor basolateral Fsk and CCH elicited any ΔI_eq_ response in the CFTR-null ileum and mid-distal colon (other segments were previously studied and were no*t*-tested this time). Then, we performed experiments in the small and large isolated intestinal mucosa of homozygous F508del mice and WT littermates (Figure 5). Ten µM of amiloride was added to the luminal bath to rule out confounding effects of ENaC inhibition by UTP [27]. One hundred µM of UTP was added, first to the luminal reservoir then to the basolateral reservoir, followed by 10 µM of Fsk. The literature suggested that the response to both luminal and basolateral UTP is mediated via P2Y4 receptors, except in the jejunum, where P2Y2 receptors may also play a role [28]. We observed a robust ΔI_eq_ to both luminal and basolateral UTP, in all segments except the jejunum, where the luminal response to UTP was weak. In all segments of the F508del homozygous mucosa, the I_eq_ response to all agonists was <10% of that in the WT mucosa. The degree of response corresponds to the availability of fully glycosylated band C CFTR protein found in the intestinal mucosa of F508del homozygous mice on the FVB/N genetic background, which is around 7% of the band C in WT mucosa [16]. Thus, we obtained no indication of a non-CFTR mediated anion secretory current in the intestinal mucosa of these mice. Appendix A shows the changes in potential difference (dPO) and electrical resistance during the experiments. The agonist-induced resistance changes were smaller than they were in the cultured colonoids, and were smaller in the ileum and proximal colon than they were in the mid-distal colon. This was due to the relative higher “tightness” of the paracellular pathway. In an epithelium with high resistance, the opening of apical anion conductances had a much larger effect than it had in a relatively leaky epithelium. This explained the larger difference in resistance of the F508del mutant compared with that of the WT epithelium in the mid-distal colon.

### 2.5. Modeling of Two-Dimensional Rectal Organoid Monolayers with Myofibroblast Coculture

Because most of the biopsies from PwCFs were rectal suction biopsies, the system was extended to rectal myofibroblast- rectal organoid (RM–RE) coculture. Rectal biopsies from HL age-matched to the age of the PwCF were used. Figure 6 shows the gene expression profile for a variety of transporter, barrier, and proliferative marker genes in the RM-RE cocultures in comparison to differentiated rectal organoid monolayers after 4 days in the respective culture medium. As previously described for CM–CE cocultures [18], strong decreases of the stem cell, pluripotency and ribosome biogenesis, and proliferative genes (LGR5, WDR43, Ki-67), and the transporter genes for anion secretion (*NKCC1*, *CFTR*, *TMEM16a*, *TMEM16F*, *CLDN2*), were observed during differentiation, as well as strong increases of the goblet cell (*MUC2*), absorptive enterocyte (*NHE3, SLC26A3, ENaC*), and barrier markers (*ZO-1*, *OCLN*, *CLDN3*, *4*, *7*, *8*). Please see Appendix A for the full names of the abbreviated gene descriptions.

### 2.6. Electrophysiological Experiments in RM–RE Monolayers from HL and Patients with Cystic Fibrosis

The ΔI_eq_ response to luminal UTP, basolateral Fsk + IBMX, and CCH, in the same concentrations as in CM–CE colonoids and in murine intestine, with and without preincubation with Ani9 (tested from 10 µM to 30 µM—in our images 30 µM were used), was investigated in RM–RE cocultures. The rectal organoids from healthy volunteers displayed a robust ΔI_eq_ in response to UTP, Fsk + IBMX, and CCH (Figure 7A). Preincubation with Ani9 did not significantly change the response to UTP, significantly reduced the response to Fsk, and increased the response to CCH. The same experiments were performed in RM–RE monolayers from PwCF. The trace shown below (Figure 7B) was obtained from five different monolayer cultures from a patient with homozygous F508del mutation, but the results were qualitatively similar in all selected CF patient monolayers. The experiments were performed with and without Ani9 preincubation. As shown in Figure 7B, the overall ΔI_eq_ response was <fifty-fold (UTP) to four-hundred-fold (CCH) less than in the RM–RE monolayers from rectal biopsies of HL. Given the lack of response to bumetanide, it is not clear if the minimal agonist-induced ΔI_eq_ truly reflected the opening of apical anion channels. The addition of Ani9 did not significantly affect the ΔI_eq_ to any agonist. The TEER values before the addition of agonists were similar in RM–RE coculture monolayers of healthy control and PwCF (Appendix A). We measured a panel of genes to compare rectal organoid monolayers from healthy control and PwCF (see Appendix A)—the TMEM16a mRMA expressions from PwCF and HL were not significantly different in RM–RE monolayers. 

The I_eq_ response of self-differentiating Caco2 cells to purinergic agonists was described as differing with time in culture, possibly suggesting increased receptor expression with enterocyte differentiation [29]. Therefore, we performed experiments in differentiated rectal organoids from HL and PwCF in the absence of and in the presence of Ani9 (Figure 8A,B and Appendix A). In differentiated rectal organoids from HL, the response to UTP decreased by about 50% in comparison with that in nondifferentiated RM–RE monolayers, but the response to Fsk + IBMX decreased by >90%, and the response to CCH decreased by approximately 50-fold. Ani9 had no inhibitory effect on any response to agonists. Interestingly, the differentiated CF rectal monolayers had a higher response to UTP than that of the nondifferentiated CF RM–RE coculture monolayers. No ΔI_eq_ to Fsk + IBMX was observed, and a very small ΔI_eq_ to CCH was observed. Since the NKCC1 inhibitor bumetanide did not have an effect on the I_eq_, in contrast to the situation in the differentiated rectal organoids from HL, it is doubtful that the I_eq_ responses to UTP and CCH represented changes in flux through a Cl^-^ conductance.

Immunocytochemical staining of TMEM16a was performed in both 3D and 2D colonic and rectal organoids (Figure 9). While the TMEM16a staining was not detected in the apical membrane, the staining pattern was different in the different conditions. In 3D cultures from healthy rectal organoids, a prominent signal was observed close to the basolateral membrane (Figure 9A). This was less intense in differentiated 3D rectal organoids, as well as in the colonic organoids, consistent with the TMEM16a mRNA expression levels. In the CF 3D nondifferentiated organoids, a subapical staining was observed, in addition to the signal near the basolateral membrane. The 2D monolayers derived from coculture with myofibroblasts displayed a more intense TMEM16a staining than that of the monolayers grown in expansion or differentiation medium for the same time span. The CF RM–RE monolayers displayed the most intense staining.

## 3. Discussion

We previously found no evidence for a CFTR-independent Ca^2+^-stimulated apical anion conductance in murine intestine, despite a robust expression of TMEM16a in murine colonocytes (Fang Xiao and Brigitte Riederer, unpublished results). In contrast to the small intestine or the large intestine, the trachea or gallbladder of CFTR-null mice displayed a marked response to purinergic or cholinergic agonists. Human intestinal cell lines, all derived from colonic tumors, also displayed strong short-circuit current responses to a Ca^2+^-dependent agonist. These findings, repeated in many laboratories, kindled the hope that agonists that elicit a fluid and secretory response via non-CFTR channels may alleviate the disease manifestations of patients with cystic fibrosis in the different epithelia affected by the disease [7]. Because of its recent cloning and characterization as a Ca^2+-^dependent anion channel [30], the focus was on TMEM16a since its discovery [31]. Now, since patients with severe CF mostly survive into adulthood because of better treatment options for their recurrent pulmonary infections and the ensuing lung destruction, the intestinal manifestations of cystic fibrosis have recently moved into the focus [32,33,34,35]. As mentioned in the introduction, the data obtained in mice, in human intestinal cell lines of cancer origin, and, to a very small extent, in patient biopsies, are controversial [8,9,10,11,12,13,14]. The genetic deletion of TMEM16a in murine intestine compromised intestinal secretory function, suggesting a regulatory role of TMEM16a on colonic anion secretion [9]. However, can TMEM16a function as an apical chloride channel in the colon? In a recent publication, nasal organoids were generated from PwCF, with nonfunctional CFTR protein tested for the presence of pathways for non-CFTR fluid secretion [36]. In that study, UTP elicited ΔI_eq_ in two-dimensional nasal cultures that was partially inhibited by Ani9 in a TMEM16a-dependent fashion, suggesting that in these cells, TMEM16a may indeed provide an apical anion conductance, albeit a small one. In our study, we wanted to specifically address the question of whether the TMEM16a anion channel, which is robustly expressed in human and murine colon, may also act as an apical anion conductance in the presence or absence of a functioning CFTR channel. For this purpose, we cultured human colonic enteroids from the transverse colon and from the rectum of HLs and from patients with severe functional defects in the *CFTR* gene as two-dimensional monolayers. Because we already knew that, in the intestine, non-CFTR pathways for increasing the luminal alkalinity exist, namely via substances that inhibit proton secretion by the electroneutral Na^+^/H^+^ exchanger NHE3 [37,38], this study focused specifically on alternative anion conductances.

The methodological approach of culturing intestinal organoids in different states of differentiation has the advantage that one can better dissect the different electrogenic ion transport pathways that are expressed along the crypt-surface axis. In contrast, in the stripped intestinal mucosa, an agonist that mediates the activation or inhibition of a variety of ion conductances in cells at the cryptal base and in the surface epithelium will elicit a compound signal that can be difficult to interpret [39]. Therefore, we assessed the agonist-induced ΔI_eq_ to the purinergic agonist UTP, the adenylate cyclase activator forskolin, and the cholinergic agonist carbachol, in both the presence and absence of different concentrations of Ani9, in both nondifferentiated and differentiated colonic and rectal organoid monolayers. Because the glycocalyx of intestinal cells creates a diffusion barrier, as does the transwell membrane whose pores may fill with extracellular matrix, we used relatively high Ani9 concentrations, from 10–30 µM. Even high concentrations of Ani9 did not significantly inhibit the UTP-induced ∆I_eq_, whereas a significant reduction in the Fsk-induced response was seen. Curiously, the CCH-induced ∆I_eq_ was significantly enhanced by Ani9 preincubation. While the data are consistent with a modulatory role of TMEM16a on Fsk-activated CFTR anion conductance, a nonspecific effect of Ani9 cannot be ruled out.

As mentioned, we were not able to induce a Fsk, CCH, or heat-stable *Eschericia coli* enterotoxin (STa)-mediated ∆I_sc_ in the small and large intestine of mice with zero expression of a functional CFTR protein (CFTR-null mice [15,40]). It is unclear, however, how similar the anion transport pathways are between murine and human intestine. Therefore, we attempted to find a human “CFTR knockout model” for the intestine. Rectal suction biopsies were obtained from patients with cystic fibrosis who participated in the Trikafta trial and had a virtual loss of CFTR function, as evidenced by electrophysiological studies of their rectal biopsies [41]. Organoids from these patients were generated, as well as organoids from rectal tissue of healthy age-matched controls, and these were grown as cocultures with rectal myofibroblasts at the bottom of the culture dish and the rectal epithelial cells on the transwell filter. As previously shown for cocultures for which both the myofibroblasts and the colonic epithelial cells were harvested from the transverse colon [18], the advantage of the coculture system is a rapid establishment of a high electrical resistance, but a prevention of differentiation. Thus, the monolayers can be compared perfectly to the differentiated monolayers, which also display high electrical resistance but an increase in absorptive transporters and a decrease of the anion secretory machinery.

We also performed Ussing chamber studies in CFTR-null and in F508del mutant murine-isolated intestinal mucosa. We used the CFTR-null mice because we wanted to evaluate the effect of the purinergic, the cAMP-dependent, and the cholinergic response in the different segments of the ileocolon. Since no ΔI_sc_ was elicited, as previously published, we did not show these results graphically. F508del mutant mice were studied to compare the agonist-induced response pattern with that seen in CF organoids that were homozygous for F508del. We know that the F508del mice on the FVB/N background express approximately 5–7% of the mature band C CFTR protein in their brush border membrane [15]. Therefore, it is easy to attribute the difference in ΔI_sc_ between the CFTR-null and the F508del epithelium to the activity of the residual CFTR anion conductance. This approach proved useful when interpreting the ΔI_sc_ pattern in the RM–RE and the differentiated monolayers from the PwCFs, who were either homozygous or compound heterozygous for F508del. In contrast to the ΔI_sc_ response pattern in the mice, which was similar, albeit much reduced, to that observed in WT epithelium (with Fsk eliciting the strongest ΔI_sc_), this was completely different in the RM–RE as well as the differentiated monolayers from the response pattern in the studied CF-patients. In both RM–RE and differentiated monolayers, the UTP-induced ΔI_eq_ was relatively highest, no response to Fsk was observed, bumetanide had no effect, and Ani9 did not inhibit the UTP-induced ΔI_eq_, in contrast to the effect of Ani9 in the nasal organoid-derived monolayers from PwCF [36]. Therefore, we considered it unlikely that the UTP-induced ΔI_eq_ represents TMEM16a activity. The fact that it is somewhat larger in the DMs than in the RM–RE monolayers is likely due to a higher expression of purinergic receptors in the latter.

Consistent with the absence of an anion secretory signal with a TMEM16a signature, TMEM16a was also not present in the apical membrane, as observed by the use of a mixture of three custom-made and a commercially available anti-TMEM16a antibodies (see the Methods section for details). The TMEM16a staining pattern in the CM–CE monolayers was different with the commercially available and the custom-made anti-TMEM16a antibody, but both of them stained intracellular organellar structures, which did not colocalize with the F-actin stain phalloidin (Figure 2). We then used the custom-made anti-TMEM16a antibody to stain 3D transverse colonic and rectal organoids in the nondifferentiated and differentiated states (Figure 9), as well as CM–CE and RM–RE cocultures. While the TMEM16a staining was not detected in the apical membrane, the staining pattern was different in the different conditions. In 3D cultures from healthy colonic and moreso from healthy rectal organoids, a prominent signal was observed close to the basolateral membrane. This was less intense in differentiated 3D rectal and colonic organoids, consistent with the TMEM16a mRNA expression levels. In the CF 3D nondifferentiated organoids, a subapical staining was observed, in addition to the signal near the basolateral membrane, which was also observed in differentiated CF 3D organoids. The staining pattern near the basal enterocyte pole resembles the pattern described for the murine colon [9]. We did occasionally observe cells that may have been goblet cells, which displayed an apical staining with TMEM16a, but this will require further study. The staining pattern of 2D monolayers from the rectal organoids was similar to that seen in Figure 2 and discussed above.

Another important question is whether agents other than purinergic, cholinergic, and cyclic nucleotides can be identified, which bring the subapical TMEM16a into the apical membrane in the CF organoids. In the airways, where immunohistochemical staining of TMEM16a in the apical membrane has been detected, the incubation with Th2 cytokines, such as IL4 or IL13, strongly upregulates the expression and apical conductance of TMEM16a. Of course, Th2 cytokines, which play a major role in the pathogenesis of ulcerative colitis, would not be a treatment option to improve gut fluidity. Nevertheless, the study of these cytokines in the organoid system may be interesting in the context of the role of TMEM16a in inflammatory diarrhea. We used E-type prostaglandins in some of our culture media, but while they impede differentiation, they do not alter the TMEM16a localization. All in all, we believe that this culture system is ideal for testing selected strategies to improve anion conductance in PwCFs.

Finally, how do our data fit into the published landscape regarding intestinal TMEM16a function? Clearly, the Ani9-mediated inhibition of TMEM16a does not result in the same alterations in the agonist-induced ΔI_sc_ as described for the intestine-specific TMEM16a-knockout mouse, which also showed a decrease of the CCH-induced ΔI_sc_ [9]. This may be explained by a change in the differentiation pattern of the intestine-specific TMEM16a knockout mouse The histological images of the villin promotor-dependent TMEM16a knockout epithelium appear to be different from those from WT mucosa [20,42]. It has been known for a long time that the genetic deletion of a variety of ion transporters results in severe alterations of the epithelial morphology, and recent studies in intestinal organoids from genetically altered mice confirm that the lack of CFTR, or the lack of NHE2, results in changes in the proliferative pattern or in a shift in the lineage differentiation pattern toward the secretory lineage (goblet, tufts, and enteroendocrine cells) [43,44]. Given the intricate but strong involvement of TMEM16a in cellular proliferation, an effect of the intestinal *Tmem16a* deletion on epithelial differentiation, with a resultant alteration of the response pattern to CFTR-agonists, is feasible [45].

We believe that our approach offers unique advantages over other experimental approaches in studying ion channel function in human colonic epithelium, such as freshly excised biopsies in Ussing chamber setups [46] or the 3D organoid swelling assay [16]. The freshly isolated biopsies need to be inserted into a tissue holder with a very small aperture, which results in high fluid resistances, drift of the potential difference, and the danger of edge-damage-induced false “secretory currents”. In contrast, our monolayers are grown on filters with much larger diameters and the development of electrical resistance of the monolayer can be monitored daily, before the start of the electrophysiological measurements. Secondly, our technique allows the separation of the proliferative, nondifferentiated epithelial cells resembling the transit amplifying cells in the lower cryptal neck zone, which highly express the anion ion transport machinery, such as CFTR, TMEM16a and f, NKCC1, Claudin2 (for paracellular permeation of the cations and water that needs to follow the anion secreted via the apical membrane), AE2, and NBCn1, from the differentiated colonocytes resembling the cryptal mouth zone. The latter express the electrolyte and fluid-absorptive machinery NHE3, SLC26A3, ENaC, the BK channels subunits, the “sealing” claudins, and the goblet cell markers MUC2 and BK channel subunit LRRC26. Therefore, the negative deflections of the potential difference, which indicate the opening of anion channels, are not partially compensated by the positive deflections caused by luminal K^+^ channels or by the negative deflections of ENaC inhibition, all of which are activated/inhibited by the same agonists as the anion channels. This advantage can be clearly seen in our CF monolayers, in which the tiny I_eq_ and PD deflections are all in the negative range. Thirdly, a combination of the electrophysiological measurements with dual excitation fluorometric dye methods allows the assessment of both electrogenic ion transport processes and electroneutral ones [18] that may influence the juxtamucosal pH [47] and assess fluid absorption/secretion [48]. Thus, we believe that our method is of excellent use for personalized medicine and for the assessment of individual functional alterations of genetically affected ion transporters.

In conclusion, this study found that the Ca^2+^-activated anion channel TMEM16a is highly expressed in human CM–CE cocultures resembling transit amplifying cells of the colonic cryptal neck zone, both from HL and from PwCF. While TMEM16a may play a role in modulating agonist-induced CFTR-mediated anion currents, it is not localized in the apical membrane and it does not function as an apical anion channel in CF and healthy human colonic epithelium. Because early electrophysiological studies in rectal biopsies of patients with homozygous mutation in the *CFTR* gene indicated that anion conductances that did not exhibit a CFTR-typical inhibitor profile may result in a clinically beneficial phenotype [49], a continuing search for the molecular nature of non-CFTR non-TMEM16a anion conductances in the gut seems to be warranted.

## 4. Material and Methods

### 4.1. Human Subjects

The mutational status of the PwCFs in this study were previously determined, and the patients were screened for participation in the TRIKAFTA trial. The results of that study were published [41,50]. The healthy subject biopsies were collected from volunteers without CF or intestinal disease. The patient characteristics of those patients who donated a biopsy for organoid culture are stated in Appendix A. The patients were informed of the purpose of the study and consented before the extra biopsy for organoid establishment was taken. The study was conducted according to the guidelines of the Declaration of Helsinki, and approved by the Institutional Review Board of Hannover Medical School (Nr. 8535_BO_K_2019., Nr. 8922_BO_S_2020). The near-complete loss of CFTR function in the rectal mucosa of the patients was been assessed by conventional electrophysiological techniques [41].

### 4.2. Mice

All experiments involving animals were approved by the Hannover Medical School Committee and an independent committee assembled by the local government. The application and permission numbers were Az. 33.14-42502-04-14/1549 and Az 33.12-42502-04-19/3197 for breeding “stressed strains”. Experiments were performed with *Cftr* wild-type (WT) mice and with *Cftr* mice that were homozygous for the F508del mutation (*F508del*^mut/mut^) [51]. The strains had been bred for >10 generations on a congenic FVB/N background at Hannover Medical School. For each experimental animal, an age- and sex-matched littermate was raised under identical breeding conditions. Special breeding conditions were used for the *F508del*^mut/mut^ and *Cftr-null* mice, as well as for the respective controls, as previously described [52]. *F508del*^mut/mut^ and the respective WT littermates were cohoused (if feasible—males may have needed separation because of fighting), and received the same diet/drinking solution (40 Na_2_SO_4_, 75 NaHCO_3_, 10 NaCl, 10 KCl, 23 g/L PEG 4000).

### 4.3. Human Organoid Cultures and Cocultures with Myofibroblasts

Biopsies from transverse colon and rectum of healthy donors or rectum of patients with cystic fibrosis were collected by endoscopy after ethics approval and informed consent and used to establish 3D organoid cultures, according to Sato et al. [53], with minor modifications, as described in [18]. Briefly, biopsies were cut into smaller pieces, washed with cold PBS, and incubated with 10 mM EDTA for 1 h. Then, crypts were mechanically isolated and plated in 50% Matrigel (Corning, Kaiserslautern, Germany; before 2021) or in 40% Cultrex (Bio-Techne, Wiesbaden, Germany; from 2021 and thereafter) supplemented with 1 μM Jagged-1 peptide. After polymerization, expansion media were added to the culture well. For details, please see the Appendix A.

Colonic resectates and rectal biopsies were used to isolate intestinal myofibroblasts according to the methods of Khalil et al. [54], as previously described [18]. Myofibroblasts were maintained in EMEM medium (Lonza, Cologne, Germany) containing 10% FBS, penicillin/streptomycin, and 1% of non-essential amino acids and used between passages 6 to 12. Two days before coculture, myofibroblast cells were seeded in a 24-well plate (∼4–7 × 10^4^/well) at 37 °C, with 5% CO_2_.

The 3D organoids were dissociated and cultured on transwell inserts (6.5 mm diameter polyester membrane with 3 or 0.4 µM pores; Corning, Kaiserslautern, Germany) to form monolayers. After reaching confluency in EM, the culture inserts were either transferred to a 24-well plate containing confluent myofibroblast culture on the bottom of the wells for coculture condition, exposed to differentiation medium for differentiation condition, or maintained in EM for non-differentiated condition, as previously described [18], and used for analysis after 4 days.

### 4.4. Immunocytochemical Localization of TMEM16a and CFTR in Intestinal 2D and 3D Cultures

Organoids monolayers grown on transwell inserts were washed with PBS, fixed in 2% PFA in PBS for 30 min at room temperature, consequently washed with PBS, and blocked/permeabilized in PBS containing 5% normal goat serum (Thermo Scientific, Darmstadt, Germany) and 0.2% Triton X-100 for 30 min at room temperature. Samples were exposed to primary antibodies overnight at 4 °C and to secondary antibodies for 1 h at room temperature. Antibody mixtures were prepared in PBS containing 5% normal goat serum and 0.1% Triton X-100. After antibody incubations, wash steps were performed in PBS plus 0.1 Triton X-100. Filters were excised from the insert and mounted with ImmunoSelect^®^ Antifading Mounting Medium (Dianova, Germany). One layer of double-sided adhesive tape (Tesa, 05338) was used as a spacer between the slide and the cover slip. Whole-mount human 3D colonoids and rectal organoids were prepared, as described by Dekkers et al. [55].

Three custom-made polyclonal antibodies generated against DREEYVKRKQRYEVDFNLE, FEEEEDHPRAEYEARVLEKSLR and MEECAPGGCLMELCIQL peptides [24], respectively, named as Ano1-132, Ano1-133, and Ano1-134, were collectively used (1:200 dilution each) to detect TMEM16a. For comparison, TMEM16a was also detected with recombinant rabbit anti-TMEM16a [SP31] (1:200, Abcam, Cambridge, UK). CFTR was detected with mouse anti-CFTR570 antibody (1:200) [56]. Secondary antibodies included goat anti-mouse Alexa Fluor 488 and goat anti-rabbit Alexa Fluor 568 (1:500, Invitrogen, Darmstadt, Germany). Rhodamine conjugated Ulex Europaeus Agglutinin I (UEA I) (1:500, Vector Laboratories) to stain mucus [57], DAPI to stain nucleus, and Phalloidin-iFluor 647 Reagent (Abcam) to stain F-actin were optionally used, in combination with secondary antibodies. Images were obtained by a TCS SP8 confocal fluorescence microscope (LEICA Microsystems, Wetzlar, Germany) and analyzed by LAS X (LEICA Microsystems), Fiji [58], and/or Imaris (Oxford Instruments, Abingdon, UK) version 8.2.1 software.

### 4.5. Electrophysiological Measurements in CM-CE and Differentiated Monolayers

Confluent monolayers of organoids grown on 0.33 cm^2^ transwell inserts with a 3 or 0.4 µM pore size (Corning, Kaiserslautern, Germany) were cut with a hot blade and mounted in the easy-mount Ussing chamber (easy-mount system, Physiologic Instruments) using the P2302T slider (Physiologic Instruments, San Diego, CA, USA). KCL-agar-filled fine tip electrodes (Physiologic Instruments, P2020-S Electrode Set) were used. The voltage electrodes were inserted bilaterally in close vicinity to the mounted epithelium, and the current electrodes were inserted with more distance. The luminal and serosal compartments were continuously perfused with buffer solutions with the following composition: luminal (in mM: 106.5 NaCl, 20 NaHCO_3_, 2.25 KCl, 1.2 MgSO_4_, 2 CaCl_2_, 2.25 KH_2_PO_4_, 10 mannitol, and 10 Na-gluconate) and basal (in mM: 106.5 NaCl, 20 NaHCO_3_, 2.25 KCl, 1.2 MgSO_4_, 2 CaCl_2_, 2.25 KH_2_PO_4_, 10 glucose, and 10 Na-pyruvate). Buffers were gassed (95% O_2_/5% CO_2_) and maintained at 37 °C to provide a pH of 7.4. After 40 min of measuring, the basal parameter different drugs were added to the chamber. These included amiloride (10 µM luminal), Ani9 (10/30 µM luminal), UTP (100 µM luminal), IBMX (100 µM, serosal), forskolin (10 µM, serosal), carbachol (100 µM serosal), and bumetanide (100 µM serosal). Experiments were conducted under open-circuit conditions. The transepithelial resistance (R) and transepithelial voltage (dPO) were recorded every 6 s using a custom-made voltage/current clamp apparatus (Klaus Mussler, Aachen, Germany). The equivalent short-circuit current (I_eq_) was determined from continuous dPO and R recordings according to Ohm’s law (where I_eq_ is equal to dPO divided by R). In order to calculate ΔI_eq_, the average response peak was subtracted from the baseline average one to two minutes before adding the drug.

### 4.6. Electrophysiological Measurements in Proximal and Mid-Distal Isolated Colonic Mucosa from F508del and WT Mice

For comparative reasons, we performed experiments in muscle-stripped and chemically denervated proximal and distal colonic mucosa to study the effect of the agonists and antagonist in *F508del*^mut/mut^ and WT littermate mucosa, as well as on *Cftr*-null mice and WT littermates. The experiments were performed as recently described for the jejunum [59]. Muscle-stripped proximal and mid-distal colonic mucosa were mounted in Ussing chambers with an exposed area of 0.126 cm^2^. To ensure a purely epithelial action, the muscle layers harboring the mesenteric plexus neurons were stripped and 1 µM tetrodotoxin was added to the serosal side to chemically denervate the tissue. The serosal solution contained (in mM) 108 NaCl, 25 NaHCO_3_, 3 KCl, 1.3 MgSO_4_, 2 CaCl_2_, 2.25 KH_2_PO_4_, 8.9 glucose, and 10 sodium pyruvate, and it was gassed with 95%O2/5%CO_2_ (pH 7.4). Tetrodotoxin (1 µM) and indomethacin (3 µM) were added serosally. The mucosal solution contained the same electrolytes, except that the glucose was replaced by mannitol and the pyruvate was replaced by NaCl. After basal parameters were measured for 30 min, the respective drugs, as indicated in the figure legends, were added to the mucosal or serosal solution. Transepithelial I_eq_ was calculated as the µEq.cm^−2^ tissue surface area. The peak electrical response was averaged over 60 sec and taken as the ∆I_eq_ response.

### 4.7. RT-qPCR Analysis

Total RNA was isolated using RNeasy Mini Kit (Qiagen, Hilden, Germany). RevertAid First Strand cDNA Synthesis Kit (Thermo Scientific) was used for reverse transcription with oligo dT primers. Quantitative PCR was performed with a Rotor Gene Q system (Qiagen), using 10ng of total cDNA 500 nM each of forward and reverse primers (Appendix A), and qPCRBIO SyGreen Mix Lo-ROX (Nippon Genetics, Düren, Germany).

### 4.8. Bioinformatic Analysis and Statistics

Multiple sequence alignment was performed with PRALINE online toolbox [60]. Data analysis and graphing were performed using Microsoft Excel 2016 and GraphPad Prism version 8.0.2 (GraphPad Software Inc., San Diego, CA, USA) software. Data are presented as mean ± SEM. Statistical significance, as determined by *p*-values from an unpaired, two-tailed parametric *t*-test, is shown as ns (not significant) *p* > 0.05, * *p* ≤ 0.05, ** *p* ≤ 0.01, or *** *p* ≤ 0.001.

## Figures and Tables

**Figure 1 ijms-24-14214-f001:**
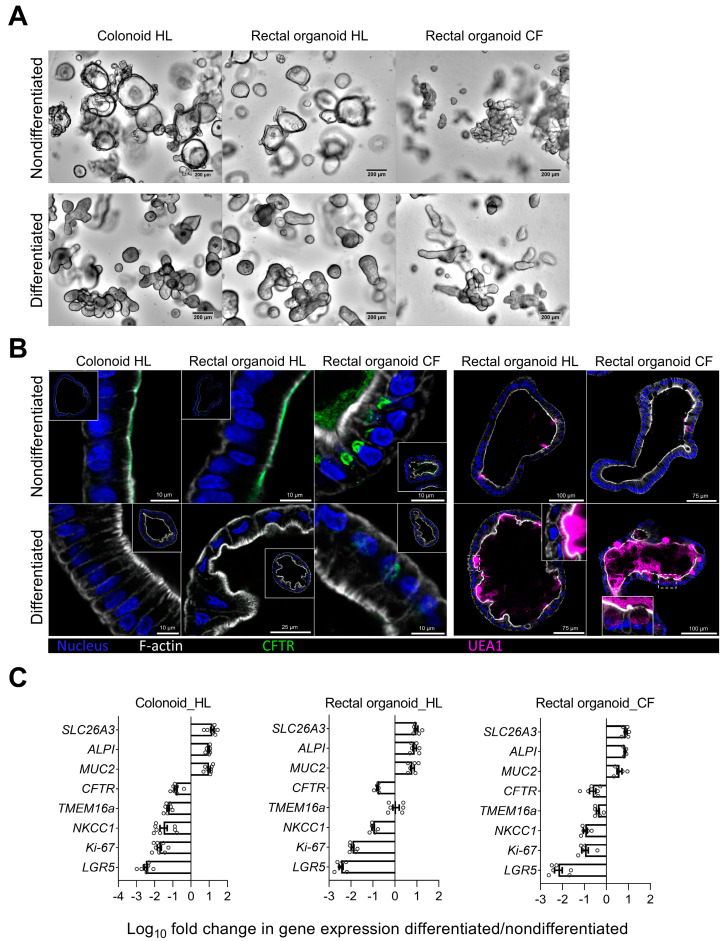
Comparison of human intestinal organoids established from healthy or CF individuals, in a nondifferentiated state where cultures were grown in expansion medium for 8 days or in a differentiation state where after initial expansion for 4 days, the cultures were maintained in differentiation medium for 4 more days. (**A**) Light microscopy images showing 3D cultures of colonoids and rectal organoids from a healthy subject (HL) and rectal organoids from a CF patient (CF). (**B**) Immunocytochemical staining of 3D organoids from the transverse colon and from the rectum of a healthy donor and a homozygous F508del patient in the nondifferentiated and differentiated states, with staining for CFTR (green) and for the goblet cell marker UEA1 (magenta), F-actin (white, detected with phalloidin- iFluor 647), and DAPI (blue, detected with DAPI). (**C**) Log_10_ fold change in mRNA expression profile between the nondifferentiated and differentiated 3D organoid cultures. Please see Appendix A for the full names of the abbreviated gene descriptions. Negative values (as for proliferative markers such as *LGR5* and *Ki-67*) indicate decrease and positive values (as for differentiation marker such as *MUC2*, *ALPI*, and *SLC26A3*) indicate increase in corresponding gene expression upon differentiation.

**Figure 2 ijms-24-14214-f002:**
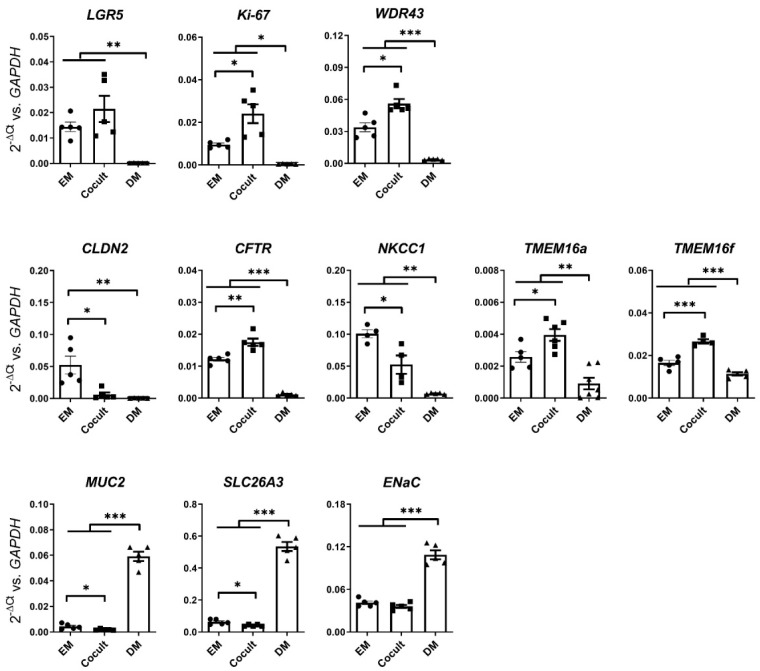
mRNA Expression profile of a selection of proliferation markers, ion transporters, and differentiation markers determined by RT-qPCR in comparison to *GAPDH* reference in nondifferentiated (EM), CM–CE cocultured (Cocult), and differentiated (DM) colonoid monolayers prepared from healthy subjects. Student’s *t*-test, * *p* ≤ 0.05, ** *p* ≤ 0.01, or *** *p* ≤ 0.001.

**Figure 3 ijms-24-14214-f003:**
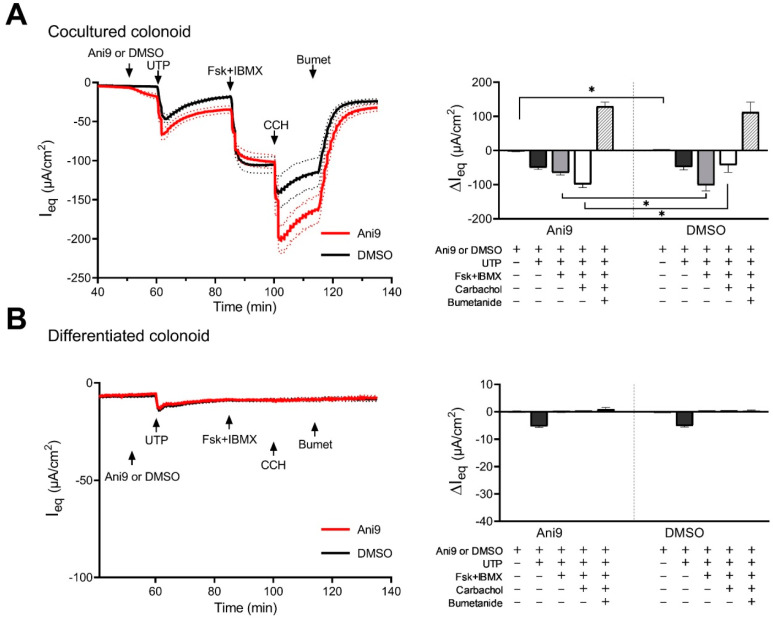
Electrophysiological analysis of anion channel activity in human CM–CE coculture (**A**) and differentiated (**B**) colonoid monolayers by the Ussing chamber approach. Cultures pretreated with amiloride (10 µM, luminal) were treated with TMEM16a-specific inhibitor Ani9 (30 µM, luminal; red trace) or DMSO vehicle control (black trace) and analyzed for the changes in the equivalent short-circuit current (ΔI_eq_) in response to 100 µM UTP (luminal), 10 µM forskolin in combination with 100 µM IBMX (Fsk + IBMX, basolateral), 100 µM carbachol (CCH, basolateral), and 100 µM bumetanide (Bumet, basolateral). Mean differences in ΔI_eq_ are calculated and presented in bar-graphs as mean ± SEM; *n* = 6; unpaired two-tailed parametric *t*-test, * *p* ≤ 0.05.

**Figure 4 ijms-24-14214-f004:**
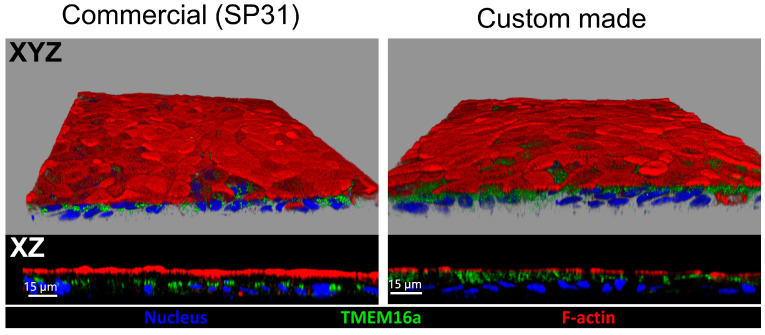
Comparison of a commercially available antibody (SP31 clone) and a combination of custom-made antibodies (Ano1-132, Ano1-133, and Ano1-134) in the detection of TMEM16a in human nondifferentiated colonoid monolayer cultures. In both sets of experiments, the signal for TMEM16a mainly revealed an intracellular organellar staining pattern and did not colocalize with the apical F-actin stained with a phalloidin conjugate. The conservation of the epitopes used to generate custom-made rabbit sera among human and mouse TMEM16a protein sequences is shown in Appendix A. Blue: nucleus (DAPI), green: TMEM16a, red: F-actin (phalloidin).

**Figure 5 ijms-24-14214-f005:**
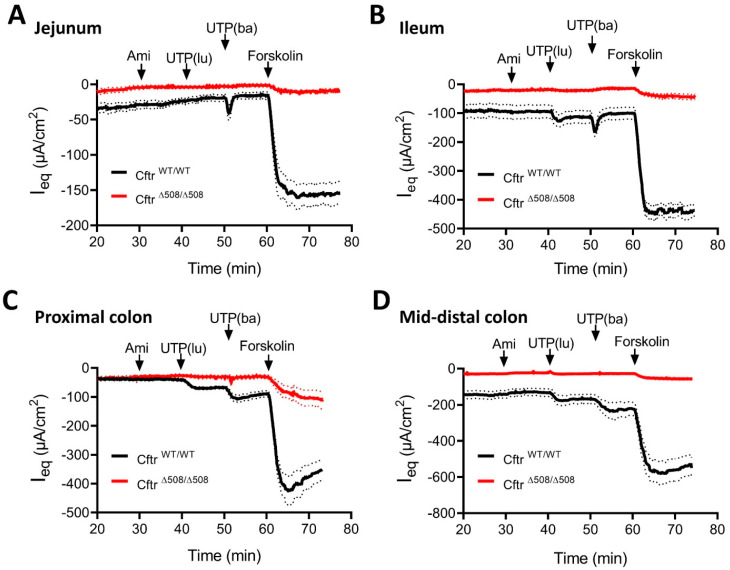
Equivalent short-circuit current response (ΔI_eq_) of isolated intestinal mucosa of homozygous F508del mice (mt, red trace) and wild-type littermates (wt, black trace). Graphs show results for (**A**) jejunum (wt *n* = 9, mt *n* = 6), (**B**) ileum (wt *n* = 8, mt *n* = 5), (**C**) proximal colon (wt *n* = 9, mt *n* = 4), and (**D**) mid-distal colon (wt *n* = 11, mt *n* = 10) segments of the mouse intestine. Samples were pretreated with amiloride (Ami, 10 µM, luminal), then, anion secretion as a function of the ΔI_eq_ response to 100 µM UTP [first luminal (lu), then with basolateral (ba)], followed by 10 µM forskolin (basolateral) was determined. In all segments, F508del homozygous mucosa presented <10% of the ΔI_eq_ response to all agonists, compared to the WT mucosa. Data for transepithelial voltage and resistance of these samples are presented in Appendix A.

**Figure 6 ijms-24-14214-f006:**
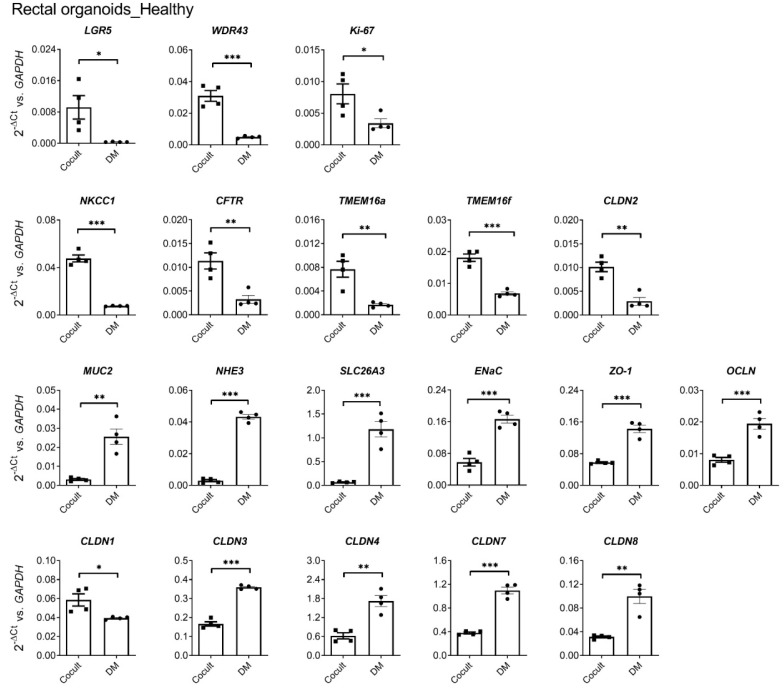
Gene expression profile for a variety of proliferative markers, ion transporters, barrier, and differentiation markers in the RM–RE cocultures (Cocult) in comparison to differentiated rectal organoid monolayers (DM). The cultures were initially grown in expansion medium to form a monolayer, then maintained in either coculture or differentiation conditions for 4 days. Student’s *t*-test, * *p* ≤ 0.05, ** *p* ≤ 0.01, or *** *p* ≤ 0.001.

**Figure 7 ijms-24-14214-f007:**
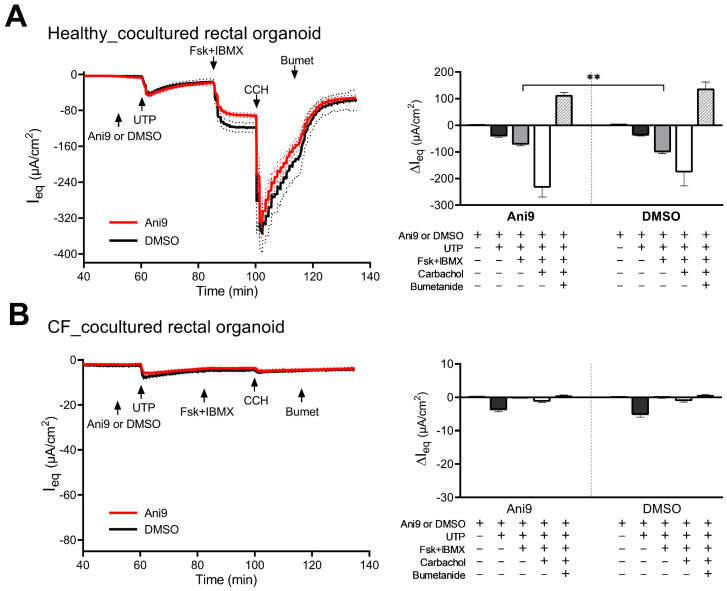
Electrophysiological characterization of anion channel activity in human RM–RE cocultures established from healthy (**A**) or CF (**B**) subjects. The cultures were pretreated with amiloride (10 µM, luminal), thereafter exposed to TMEM16a-specific inhibitor Ani9 (30 µM, luminal; red trace) or DMSO vehicle control (black trace), and analyzed for the changes in the equivalent short-circuit current (ΔI_eq_) in response to 100 µM UTP (luminal), 10 µM forskolin in combination with 100 µM IBMX (Fsk + IBMX, basolateral), 100 µM carbachol (CCH, basolateral), and 100 µM bumetanide (Bumet, basolateral). ΔI_eq_ were calculated and are presented in the bar graphs as mean ± SEM. Healthy_DMSO *n* = 5, healthy_Ani9 *n* = 11, CF_DMSO *n* = 6, CF_Ani9 *n* = 9; unpaired two-tailed parametric *t*-test, ** *p* ≤ 0.01.

**Figure 8 ijms-24-14214-f008:**
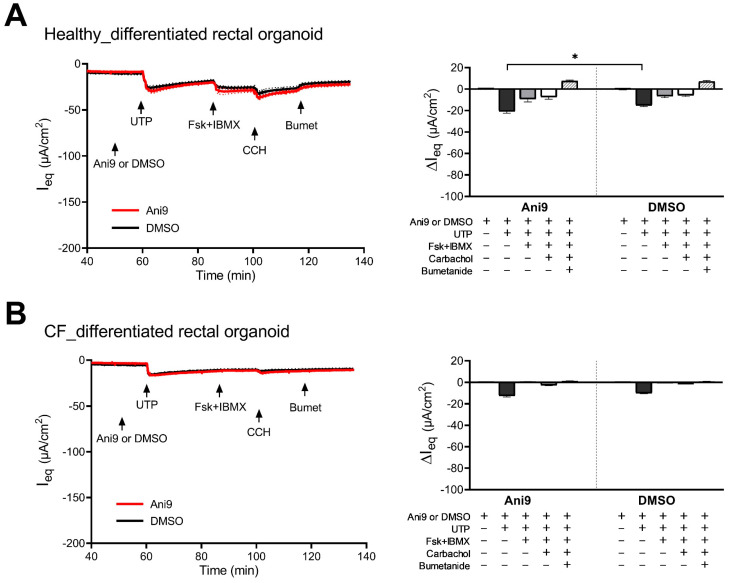
Electrophysiological characterization of anion channel activity in human differentiated rectal organoid monolayers established from healthy (**A**) or CF (**B**) subjects. The cultures were pretreated with amiloride (10 µM, luminal), thereafter exposed to TMEM16a-specific inhibitor Ani9 (30 µM, luminal; red trace) or DMSO vehicle control (black trace), and analyzed for the changes in the equivalent short-circuit current (ΔI_eq_) in response to 100 µM UTP (luminal), 10 µM forskolin in combination with 100 µM IBMX (Fsk + IBMX, basolateral), 100 µM carbachol (CCH, basolateral), and 100 µM bumetanide (Bumet, basolateral). ΔI_eq_ were calculated and are presented in the bar graphs as mean ± SEM. Healthy_DMSO *n* = 4, healthy_Ani9 *n* = 5, CF_DMSO *n* = 4, CF_Ani9 *n* = 5; unpaired two-tailed parametric *t*-test, * *p* ≤ 0.05.

**Figure 9 ijms-24-14214-f009:**
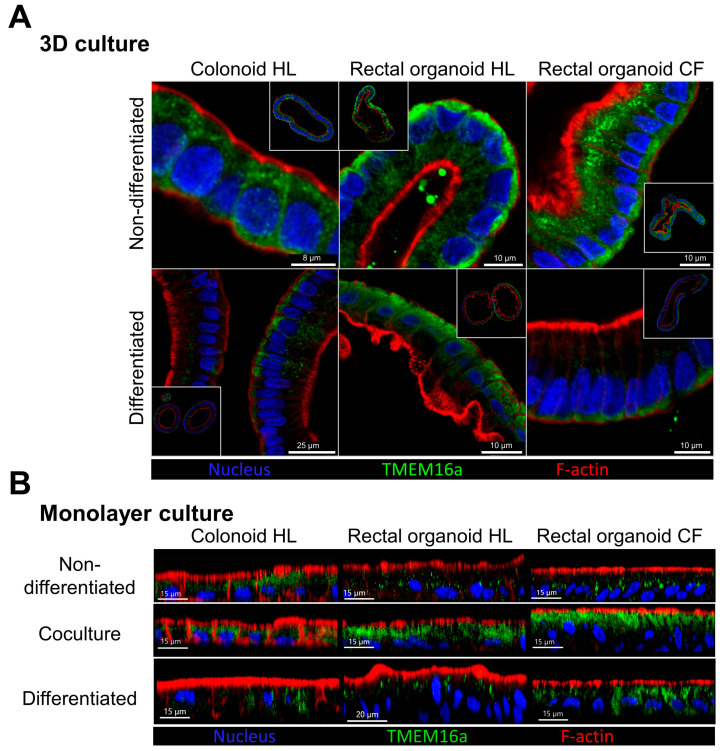
Localization of TMEM16a in colonoid and rectal organoid cultures from a healthy subject and rectal organoid cultures from a CF patient. TMEM16a was detected by immunostaining in 3D organoid cultures in nondifferentiated and differentiated states (**A**) or monolayer cultures maintained in nondifferentiated, coculture, and differentiated conditions (**B**). The signal for TMEM16a is mainly found in the intracellular compartment and does not colocalize with apical F-actin staining. Blue: nucleus (DAPI), green: TMEM16a (combination of custom-made rabbit sera against TMEM16a epitopes), red: F-actin (phalloidin conjugate).

## Data Availability

All data underlying the results are available as part of the article. The own unpublished data that was cited for the purpose of discussion are available on request from the corresponding author.

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
