# Peer review of "The Anion Channel TMEM16a/Ano1 Modulates CFTR Activity, but Does Not Function as an Apical Anion Channel in Colonic Epithelium from Cystic Fibrosis Patients and Healthy Individuals"

_ijms, 2023, doi:10.3390/ijms241814214_

Round 1

Reviewer 1 Report

This is an interesting work containing a huge amount of measured experimental data. The authors used the almost hundred-year-old Ussing method to the limit by specifically blocking all known ion channels and using channel activators. The results indicate that the calcium-activated chloride channel cannot replace the CFTR channel that is damaged or absent in cystic fibrosis. In their research, they used sophisticated co-cultures of different cells. F508del mutant and CFTR knockout (null) mice were also tested. Staining showed that the TMEM16A calcium-activated chloride channel is absent in the luminal layer of cells.

The research is precisely described. In Supplementary materials there are raw data on how the potential and resistance of the cell layer changes under the influence of various factors. I am missing only a description of the electrodes used for electrophysiological measurements. How many electrodes were there? What were the electrodes? Was a salt bridge used? Two additional sentences should be added to the Method section and paper should be published.

Author Response

Dear Reviewer, thank you for your positive opinion on our manuscript. We have now added the details about the electrodes which were used in the Ussing chamber experiments. 

Reviewer 2 Report

The functional expression of TMEM16a (ANO1) in the intestine is controversial while it may represent a therapeutic target for people with cystic fibrosis. By combining electrophysiology, immunostaining and gene expression on human and mouse intestinal mucosal tissues or cell-derived organoids in 3D- and 2D cultures, Salari and collaborators convincingly demonstrate that TMEM16a, although expressed in the colonic and rectal epithelium, is however not located at the cell plasma membrane. Thus, TMEM16a cannot function as an apical anion channel. This knowledge has important consequences for the development of molecules aimed at enhancing anion secretion in the intestine of people harboring mutations that affects CFTR channel activity. The manuscript could be improved by addressing the following points.

Major points:

1) The conclusion that TMEM16a may play a role in modulating CFTR activity is not really justified by the data presented in the manuscript (abstract, l.476-477). The authors need to explain how Ani9 can modulate the activity of CFTR while TMEM16a is not functionally expressed at the membrane. Together with the odd effects of Ani9 on cAMP-induced and CCH-induced Ieq, it would be wise to add comments on the limited specificity of this inhibitor.

2) Without reading previous publications from the group, it is not obvious to relate CFTR functional expression in non-differentiated organoids rather than differentiated organoids. If I am correct, non-differentiated organoids mimics the secretory function of the Lieberkühn crypts whereas differentiated organoids are functionally closer to the absorptive luminal epithelial cells? This concept could be introduced better in the results section, which is hard to read as it is.

3) Why the CCH-induced Ieq response is so different between Figure S1 and Figure 3?

4) Figure 3A: the authors should show whether Ani9 reduces cAMP-induced Ieq after inhibition of CFTR with I-172.

Minor points:

1) Figure 3B is not described in the Results section.

2) Figure S3 should be described in the main text of the Results section and not referred to from the Figure 5’s legend.

3) Figure 7B: if the results were similar, why are they not shown?

4) L.361-362: there are no data on heat-stable E. coli enterotoxin-mediated Isc in the Results section!

5) It was surprising to read that 2D-cultures were performed on Transwells with membrane pores of 3 um while 0.4 um pores are commonly used. Is there a specific reason for this?

English must be improved. Here is an example among many others of a sentence that is not readable (l.202-206): “In all segments of the F508del homozygous mucosa, the response to all agonists was < 10% of that in the WT mucosa, which corresponds with the presence of fully glycosylated band C CFTR protein which we found in the intestinal mucosa of F508del homozygous mice on the FVB/N genetic background, which was approx. 7% of that in WT”.

Author Response

Point by point response to Reviewer 2:
Firstly, thank you for your thoughtful comments. Below are the responses:
Major points: 
1.) Your comment is justified. However, it  was not our focus to study the role of TMEM16a as a modifier of CFTR activity. Others, among them one of the coauthors who helped us with the detection of TMEM16a protein, have extensively studied the question. They used intestine-specific TMEM16a knockout mice as well as genetic silencing, which may be a better approach than just the use of one inhibitor claimed to be the most specific one for TMEM16a at this time, but it also bears problems, as discussed in our manuscript text. 
Our focus was to find out whether TMEM16a may operate as an alternative apical anion channel in organoid-derived epithelium from humans. The technique that we used offers the advantage to study the enterocytes of the part of the colonic crypt which expressed most TMEM16a at the tran-script as well as at the protein level. The results are not what we initially had hoped for, but they seem convincing to me.
However, since the Ani9 clearly influenced the CFTR-dependent Isc-response, our results are compatible with the concept of a regulatory role of TMEM16a on CFTR-mediated anion secretion, although they do not prove it. We also cannot rule out nonspecific effects, and I now mention this cearly in the manuscript text.This could happen at several levels of the complicated and multilevel anion secretory machinery. We ourselves have generated no data that would help clarify this issue, but taking up ideas from the literature, for example a regulation of volume, pH, or the uptake or release of other ions, of intracellular organelles may affect CFTR activity, either directly by influencing its membrane trafficking, or indirectly by influencing the electrochemical driving force. 
It is also a feature of the electrophysiological assessment of anion secretory response in Ussing chamber setups that an Isc response to a previous agonist influences the response to the next one. This is not due to the fact that the epithelium cannot generate more current, but more likely the activation of intracellular events that downregulate the anion secretory event. One example to make this better understandable: If anion secretion is stimulated by heat-stable E. coli enterotoxin (Sta) or by a cGMP analogue such as 8br-cGMP or cpt-cGMP in isolated murine jejunal mucosa, a robust Isc response is measured. Forskolin, added as a second agonist, increases the Isc further. In the absence of the cGMP-dependent kinase II, which is activated by cGMP and phosphorylates CFTR, this response is strongly reduced. A subsequent response to FSK is stronger, to about the level seen in WT mucosa after the addition of both agonists. However, this does not indicate that the tissue cannot generate higher Isc, because the addition of luminal glucose, which stimulates electrogenic SGLT1-mediated Na+ glucose absorption, increases Isc even further. One possibly is the influence of phosphodiesterase activation on the anion secretory responses, but there are others.
In the case of the sequential Isc response to UTP, FSK and CCH with and without Ani9 resulted in a decrease in the Isc response to FSK and an increase in the subsequent response to CCH. To investi-gate whether this was due to the sequential addition of other agonists prior to CCH, we also reversed the order, adding first CCH. I inserted the Ieq trace for these experiments. As you see, the enhancement of the CCH response by Ani9 is seen as well. However, we cannot rule out the possibility of nonspecific effects of Ani9, and I have mentioned that in the text now. 

2.) Thank you, we do this now. 
3.) Figure 3 shows the calculated short circuit response (∆Isc, or ∆Ieq), whereas Figure S1 gives the trace for the potential difference (PD) and the electrical resistence (R). Isc is influenced by changes in both PD and R. We measure in the open circuit mode, in which the epitheial is subjected to current pulses regularly, in order to assess the electrical resistence. The strange initial reflection of the PD at the beginning of the PD response to CCH may be due to one of the current pulses having happened in just that second. But there curves shown for PD and R are exactly the same curves as used for the calculation of the Ieq shown in Figure 3. 

4.) Please allow us to not perform this experiment, because we make the paper more vulnerable to justified criticism. We do know that the CFTRinh172 inhibits FSK-induced (or in fact by any agonist) Isc well. However, it is also clear by now that this inhibitor is not specific and influences a variety of other cellular events. 
Because of this issue, we chose to obtain biopsies from patients with severe CF mutations, which were to be included in the Trikafta trial. Only rectal suction biopsies were obtained in these patients, so we could not compare the data to tose form the colonoid monolayers obtained form the transverse colon. But the advantage was that these patients were also studied by the electrophysiologic method described in citations 41 and 50 in fresh suction biopsies, and we selected patients that did not show a CFTR-typic  Isc response in those classic Mini-Ussing chamber experiments.  We therefore were able to use human “functional CFTR knockout” material, which allowed us to circumvent the problem with nonspecific effects of CFTR inhibitors. By culturing the CF organoids for several passages, any inflammatory or fibrotic alterations that may be present in the biopsy are lost. Only genetic and epigenetic colonic epithelium-specific alterations will influence the ion transporter expression and function. 

Minor points:

1.) Figure 3B is now described in the text. 
2.) Figure S3 is now described in the results text.
3.) The results were qualitatively similar, but because the changes of the Ieq were so tiny, a summation of the curves would results in more fuzziness. This is mostly due to an improvement of our methodology, and also because we had to change both our extracellular matrix (Matrigel was not available fro some time during the pandemic) and our Transwell filters. 
4.) sorry, corrected
5.) We could not buy the 0.4 µM filters for a long time. In addition, we need to use 3µM pore sizes for the fluorometric pHi-measurements,  and had learned that it is more difficult to grow the cells to confluency on the larger pore size filters, but once is is achieved, they are more adherent. That is an advantage for the Ussing-chamber experiments. 

English language: The manuscript was thoroughly revised. 

Round 2

Reviewer 2 Report

No additional comments.